# Cold induces brain region-selective cell activity-dependent lipid metabolism

Hyeonyoung Min[1], Yale Y Yang[2], Yunlei Yang[1,3,4,5]*

[1]Department of Medicine, Division of Endocrinology, Albert Einstein College of Medicine, Bronx, United States; [2]Friends Seminary, New York, United States; [3]Department of Neuroscience, Albert Einstein College of Medicine, Bronx, New York, United States; [4]Einstein-Mount Sinai Diabetes Research Center, Albert Einstein College of Medicine, Bronx, New York, United States; [5]The Fleischer Institute for Diabetes and Metabolism, Albert Einstein College of Medicine, Bronx, New York, United States

## eLife Assessment

This phenomenological study reported that cold exposure induced mRNA expression of genes related to lipid metabolism in the paraventricular nucleus of the hypothalamus (PVH). While the paper does not address cell-type specificity or the functional role of lipids in PVH, the findings might still serve as a **useful** basis for others to explore their relevance to brain responses to cold. In the revised manuscript, the authors made adequate editions, such as new immunostaining and immunoblotting of AGTL and HSL in the PVH, and pharmacological inhibition of lipid peroxidation and lipolysis. The authors also increased the sample size of some experiments and revised the text to limit their data interpretation. Thus, the reviewers considered that these studies are **solid** in conclusively describing how the PVH is reprogrammed at the level of gene expression by cold exposure.

*For correspondence:
yunlei.yang@einsteinmed.edu

**Abstract** It has been well documented that cold is an enhancer of lipid metabolism in peripheral tissues, yet its effect on central nervous system lipid dynamics is underexplored. It is well recognized that cold acclimations enhance adipocyte functions, including white adipose tissue lipid lipolysis and beiging, and brown adipose tissue thermogenesis in mammals. However, it remains unclear whether and how lipid metabolism in the brain is also under the control of ambient temperature. Here, we show that cold exposure predominantly increases the expressions of the lipid lipolysis genes and proteins within the paraventricular nucleus of the hypothalamus (PVH) in male mice. Mechanistically, by using innovatively combined brain-region selective pharmacology and in vivo time-lapse photometry monitoring of lipid metabolism, we find that cold activates cells within the PVH and pharmacological inactivation of cells blunts cold-induced effects on lipid peroxidation, accumulation of lipid droplets, and lipid lipolysis in the PVH. Together, these findings suggest that PVH lipid metabolism is cold sensitive and integral to cold-induced broader regulatory responses.

## Introduction

Lipid metabolism in peripheral tissues has been extensively studied, including white adipose tissue lipolysis (*Grabner et al., 2021*), beiging (*Bartelt and Heeren, 2014*), and brown adipose tissue thermogenesis *Carpentier et al., 2023*; however, an essential but poorly understood element is that of lipids in the central nervous system, particularly in the hypothalamus which plays a crucial role in the regulation of systematic energy metabolism (*Waterson and Horvath, 2015*) and glucose homeostasis (*Pozo and Claret, 2018*). The brain is highly thermal sensitive in that 1°C or less temperature

changes can lead to functional alterations of the central nervous system (**Brooks, 1983**; **Wang et al., 2014**). Lipids provide a major source of energy and heat in the body, however, the mechanism underlying brain lipid metabolism regulations remains a mystery. There is evidence to show that the brain's energy metabolism largely depends on the temperature (**Guyton and Hall, 2006**; **Yu et al., 2012**). We therefore assume that cold modulates brain lipid metabolism in brain regions that express the genes encoding lipid metabolic enzymes sensitive to ambient temperature. Identifying the brain regions and genes modulated by ambient temperature is of significance in our understandings of the mechanisms of maintaining brain lipid metabolism homeostasis.

Here, we combine cold exposure and brain region-selective genetic, molecular, and lipid metabolic assays, coupled with in vivo real-time two-color fiber photometry monitoring of lipid metabolic activity, to define cold-sensitive brain region(s) and lipid metabolism and to elucidate the involved cellular mechanisms.

## Results

### Cold-induced brain region-selective gene expressions of lipid lipolytic markers in males

To investigate a potential for cold exposure to modulate gene expressions for lipid lipolysis and thermogenesis in the hypothalamus, mice were exposed to a cold (4°C) chamber for 4–6 hr, a more physiologically relevant condition. Immediately after the cold challenge, mouse brains were acutely extracted and sectioned in ice-cold oxygenated artificial cerebrospinal fluids (ACSFs). Hypothalamic sections that, respectively, include the paraventricular nucleus of the hypothalamus (PVH), lateral hypothalamus (LH), dorsomedial hypothalamus (DMH), ventromedial hypothalamus (VMH), and arcuate nucleus (ARC) were transferred to ACSF-containing incubators. Micro-punches of these brain regions were subsequently made for gene assays of lipid metabolic markers.

To evaluate the effects of cold on the gene markers of lipid lipolysis, we measured the mRNA levels of two key lipolytic markers adipose triglyceride lipase (*Pnpla2*) and hormone-sensitive lipase (*Lipe*). We observed that cold-challenged male mice showed a significant increase in the gene expressions of the both of *Pnpla2* (**Figure 1A$_1$**) and *Lipe* (**Figure 1A$_2$**) selectively in the PVH but not in other brain regions (**Figure 1B–E**). We also assayed the gene expressions of thermogenic marker uncoupling protein 2 (*Ucp2*) and additional thermogenic factors (*Cidea*, *Prdm16*). Cold did not significantly affect mRNA expressions of these thermogenic markers in all the examined regions in this study (**Figure 1**). Cold did not significantly affect gene expressions of these lipolytic and thermogenic markers in female mice (**Figure 1—figure supplement 1**). These results suggest that cold exposure (4–6 hr) could induce a rapid lipolytic activity to release fatty acids primarily in the PVH in males. Because cold did not significantly affect lipolytic markers in other regions and did not affect thermogenic markers, we next focused on studying cold-induced lipid mobilization and lipolysis in the PVH in male mice.

### Cold increases the expressions of lipid lipolytic enzymes in the PVH

To verify the cold-induced effects on lipolytic gene expressions, we next stained the protein expressions of ATGL and HSL in PVH sections using antibodies against the ATGL and HSL, respectively. Matching with the gene expression results, cold-challenged mice showed significant increases in ATGL (**Figure 2**) and HSL (**Figure 3**) expressions in both neurons and astrocytes, respectively, compared to control mice.

Moreover, we examined whether and how cold modified the phosphorylation of the HSL, a key enzyme in the regulation of lipid lipolysis. After the cold exposure of cohort groups of mice, micro-punches of PVH sections were collected and directly used for western blots of phosphorylated HSL (p-HSL), HSL, and actin. Cold significantly increased activation (S660) site phosphorylation of the HSL compared to controls (**Figure 4**).

### Cold-induced astrocytic accumulations of lipid droplets

It is increasingly appreciated that lipid droplets (LDs), endoplasmic reticulum-derived intracellular neutral lipid storage dynamic organelles (**Olzmann and Carvalho, 2019**), are predominantly accumulated in adipose tissues and liver under normal physiological conditions (**Murphy, 2001**; **Farese and Walther, 2009**). Also, there is evidence to indicate that glial cells such as astrocytes and tanycytes

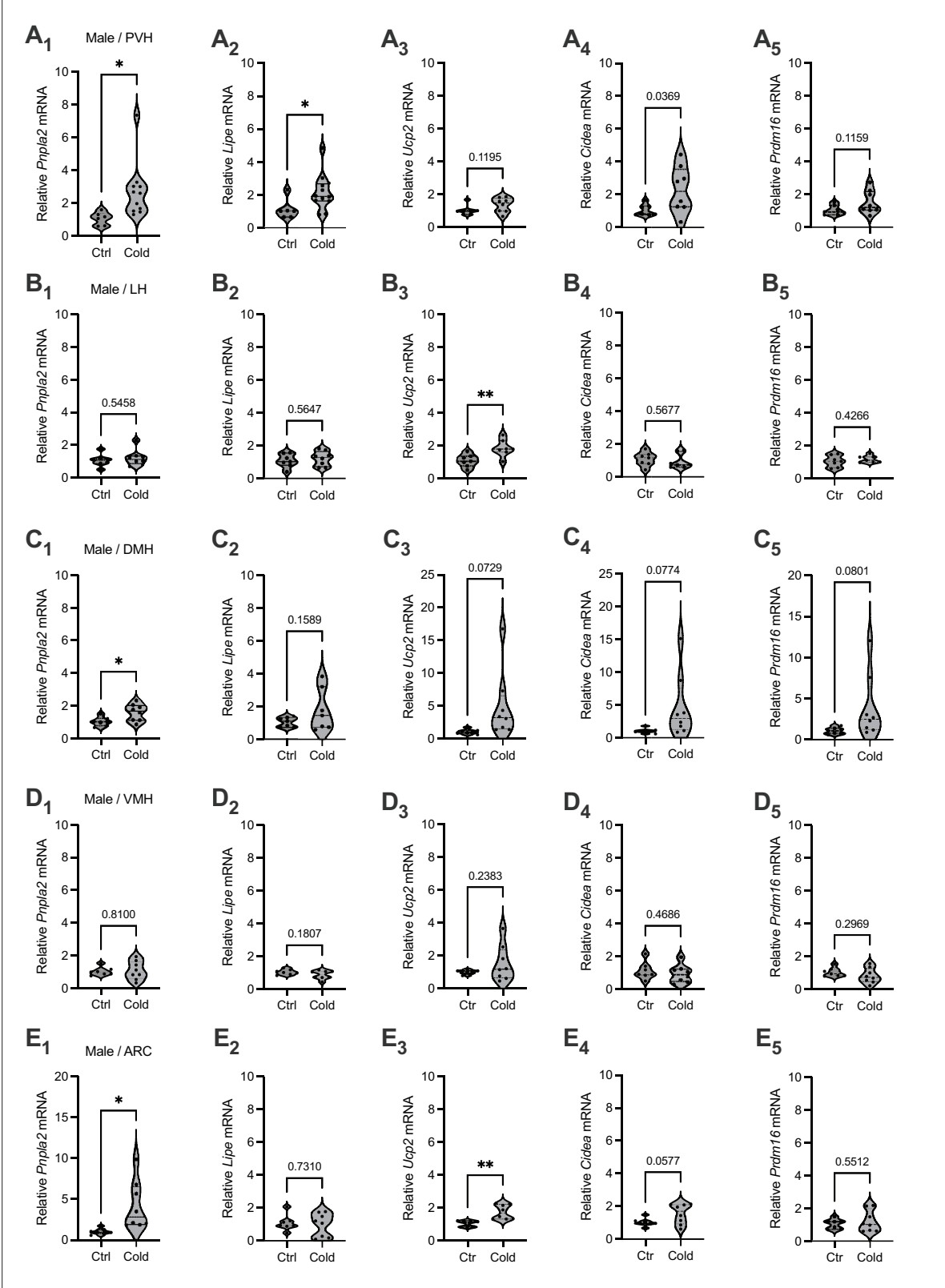

**Figure 1.** Brain region-selective responses to cold. Micro-punches of paraventricular nucleus of the hypothalamus (PVH), lateral hypothalamus (LH), dorsomedial hypothalamus (DMH), ventromedial hypothalamus (VMH), and arcuate nucleus (ARC) were made from male mice exposed to a cold chamber for 4–6 hr, which were directly used for real-time qPCR (RT-qPCR) of the gene markers of lipolysis and thermogenesis. Group data of the lipolytic marker (**A₁**) *Pnpla2* (Ctrl, $n = 6$; Cold, $n = 11$) and (**A₂**) *Lipe* (Ctrl, $n = 6$; Cold, $n = 12$) as well as thermogenic marker (**A₃**) *Ucp2* (Ctrl, $n = 7$; Cold,

*Figure 1 continued on next page*

Figure 1 continued

*n* = 8), (**A**4) *Cidea* (Ctrl, *n* = 7; Cold, *n* = 8), and (**A**5) *Prdm16* (Ctrl, *n* = 7; Cold, *n* = 8) in the PVH. Group data of the lipolytic marker (**B**1) *Pnpla2* (Ctrl, *n* = 7; Cold, *n* = 8) and (**B**2) *Lipe* (Ctrl, *n* = 7; Cold, *n* = 8) as well as thermogenic marker (**B**3) *Ucp2* (*n* = 7 each group), (**B**4) *Cidea* (*n* = 7 each group), and (**B**5) *Prdm16* (*n* = 7 each group) in the LH. Group data of the lipolytic marker (**C**1) *Pnpla2* (Ctrl, *n* = 7; Cold, *n* = 8) and (**C**2) *Lipe* (*n* = 7 each group), and thermogenic marker (**C**3) *Ucp2* (Ctrl, *n* = 7; Cold, *n* = 8), (**C**4) *Cidea* (Ctrl, *n* = 7; Cold, *n* = 8), and (**C**5) *Prdm16* (Ctrl, *n* = 7; Cold, *n* = 8) in the DMH. Group data of the lipolytic marker (**D**1) *Pnpla2* (Ctrl, *n* = 7; Cold, *n* = 8) and (**D**2) *Lipe* (Ctrl, *n* = 7; Cold, *n* = 8), and thermogenic marker (**D**3) *Ucp2* (Ctrl, *n* = 7; Cold, *n* = 8), (**D**4) *Cidea* (Ctrl, *n* = 7; Cold, *n* = 8), and (**D**5) *Prdm16* (Ctrl, *n* = 7; Cold, *n* = 8) in the VMH. Group data of the lipolytic marker (**E**1) *Pnpla2* (Ctrl, *n* = 7; Cold, *n* = 8) and (**E**2) *Lipe* (Ctrl, *n* = 7; Cold, *n* = 8), and thermogenic marker (**E**3) *Ucp2* (Ctrl, *n* = 7; Cold, *n* = 8), (**E**4) *Cidea* (Ctrl, *n* = 7; Cold, *n* = 8), and (**E**5) *Prdm16* (Ctrl, *n* = 7; Cold, *n* = 8) in the ARC. Data represent mean ± SEM. Student *t* tests were performed. *p < 0.05, **p < 0.01. Each dot represents one animal in each group of all the panels.

The online version of this article includes the following source data and figure supplement(s) for figure 1:

**Source data 1.** Lipid metabolic gene markes in different hypothalamic areas in male mice.

**Figure supplement 1.** Cold did not affect gene markers in paraventricular nucleus of the hypothalamus (PVH) in female mice.

**Figure supplement 1—source data 1.** Lipid metabolic gene markes in the PVH in female mice.

in the brain accumulate LDs under metabolic and hypoxic stress (*Geller et al., 2019*; *Smolič et al., 2021*). A recent elegant study shows that hyperactive neurons release fatty acids from phospholipids to formulate LDs in the brain and activation of neurons promotes lipolysis by increasing cytoplasmic lipases (*Ioannou et al., 2019*). These findings suggest that increased cell activity and accumulated LDs probably contribute to the cold-induced increase in the lipolytic markers we observed in this study.

To probe the mechanism of the upregulation of lipolytic markers in the PVH, we evaluated the ability for cold to accumulate LDs. After the cold challenge, mouse brains were perfused, fixed, and sectioned. Hypothalamic sections containing PVH were stained using the BODIPY 493 (BD493), an LD probe which has been well developed and widely applied to measure LD number and area in fixed tissues and live cells, respectively (*Spangenburg et al., 2011*; *Long et al., 2012*; *Liu et al., 2015*; *Geller et al., 2019*; *Ioannou et al., 2019*; *Smolič et al., 2021*). Interestingly, we observed that a short-term (30 min to 1 hr) cold exposure induced an increase in the LD area in the PVH (*Figure 5A–C*). We verified LDs by using the cytoplasmic LD-binding protein perilipin-2 (*Figure 5D–F*). To define the cell type selectivity of LD accumulation induced by cold, we co-stained LDs with the BD493, astrocytes with S100b, and neurons with neuronal marker NSE, respectively, in PVH sections. We observed that cold exposure significantly increased the BD493-positive S100b-expressing astrocytes but not NSE-expressing neurons (*Figure 5—figure supplement 1*), consistent with the previous reports that LDs were primarily accumulated in glial cells (*Bailey et al., 2015*; *Liu et al., 2015*; *Marschallinger et al., 2020*; *Smolič et al., 2021*). However, a longer (4–6 hr) cold exposure reduced the number of LDs (*Figure 5—figure supplement 2*), which might be due to cold-induced liberations of fatty acids from the LDs to mitochondria for β-oxidation.

## Cold increases Fos expressions in PVH

We next examined whether and how cold modified cell activities within the PVH. Mice were placed in a cold chamber before mouse brains were perfused and fixed. PVH sections (40 μm in thickness) were stained using anti-Fos antibodies. Mice subjected to the cold challenge showed increased expressions of the cell activity indicator Fos in the PVH as compared to control mice (*Figure 6A, B*).

## In vivo real-time fiber photometry monitoring of cold-induced lipid peroxidation

As lipid peroxidation is essential in activity-dependent accumulation of LDs (*Ioannou et al., 2019*), we next evaluated the capability for cold to induce lipid peroxidation. Fiber photometry has recently been applied to detect the levels of fluorescent biosensors in vivo by us (*Chen et al., 2022*) and others (*Sun et al., 2018*; *Andersen et al., 2023*). To achieve this goal, we took advantage of the BODIPY581/591 C11 (BD-C11) ratiometric lipid peroxidation sensor, coupled with in vivo time-lapse two-color photometry monitoring approach. We developed and validated this approach to simultaneously monitor both red and green signals in one brain region through a single implanted photometry fiber connected to a two-color photometry system, for the BD-C11 sensor as it shifts its fluorescence emission peak from 590 (red) and 510 (green) nm when the sensor is oxidized. A custom-made Optical

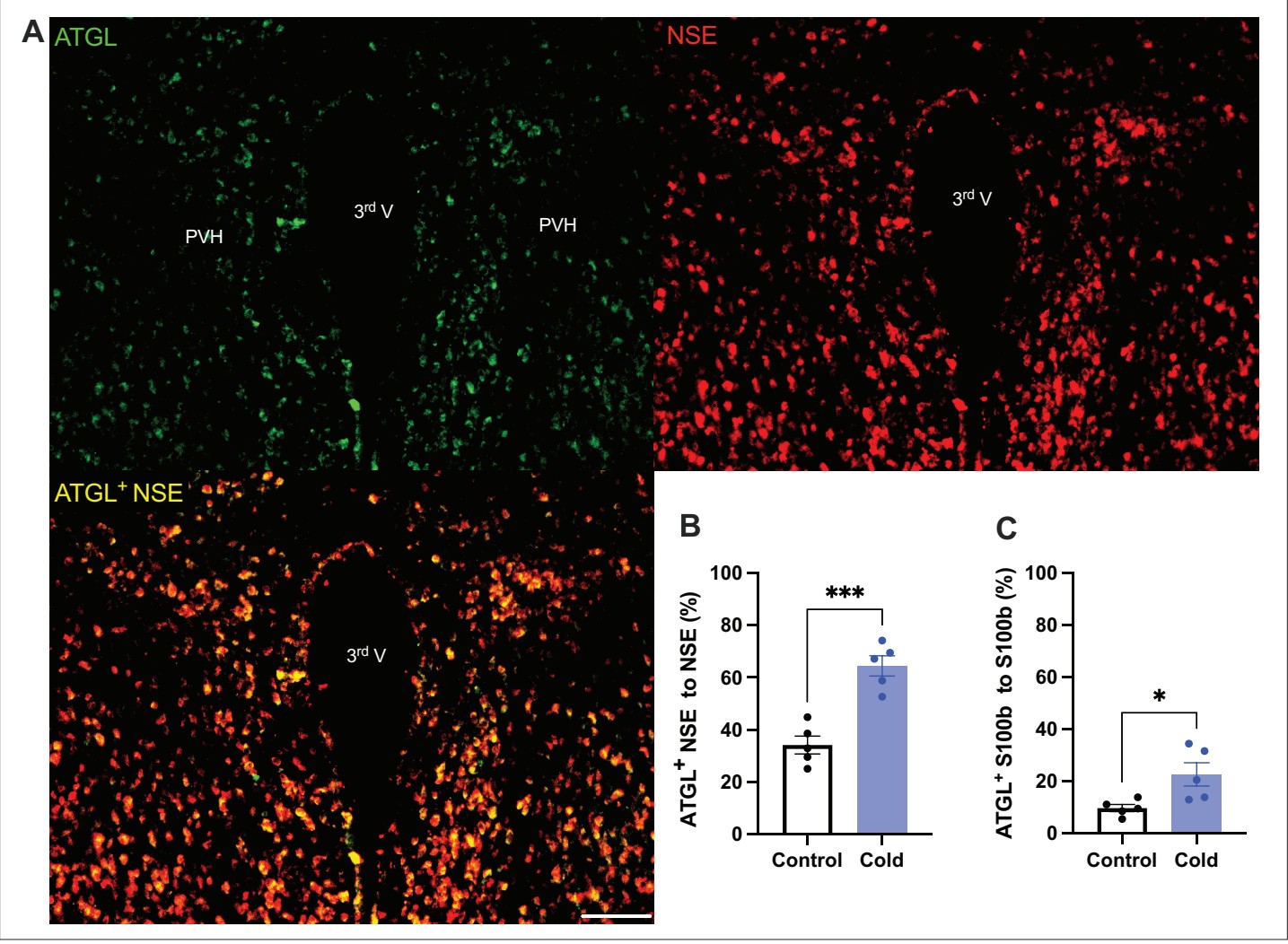

**Figure 2.** Cold increases protein expressions of ATGL in neurons and astrocytes in paraventricular nucleus of the hypothalamus (PVH). Control or cold (4–6 hr)-challenged mice were perfused and fixed. Mouse brains were sectioned. ATGL, NSE, and S100b were stained using the relevant antibodies, respectively. (**A**) Representative images of ATGL (green), NSE (red), and ATGL/NSE overlay (yellow) signals in PVH sections from cold-challenged mouse. (**B**) Group data of relative ATGL/NSE overlay signals to total NSE signals in control and cold-challenged mice ($n = 5$ each group). (**C**) Group data of relative ATGL/S100b overlay signals to total S100b signals in control and cold-challenged mice ($n = 5$ each group). Data represent mean ± SEM. Student $t$ tests were performed. ***$p < 0.001$, *$p < 0.05$. Scale bar, 100 µm for (**A**). PVH, paraventricular of hypothalamus; 3rd V, third ventricle.

The online version of this article includes the following source data for figure 2:

**Source data 1.** Group data of ATGL expressing neurons and astrocyets.

fiber <u>m</u>ultiple <u>F</u>luid injection <u>C</u>annula (OmFC) implanted over PVH was used for both BD-C11 injection and photometry monitoring of the two signals in freely behaving mice. Intra-PVH injection of the BD-C11 sensor through the OmFC was performed 4 hr prior to placing the mice in a temperature-controlled chamber. Cold potently increased the BD-C11 ratio (green to red), indicating increased lipid peroxidation (*Figure 7A*). To verify this result, we treated mice with a lipid soluble antioxidant α-tocopherol (α-TP) to inhibit lipid peroxidation. Compared to the vehicle-treated mice (*Figure 7B*), prior inhibition of lipid peroxidation with the α-TP blunted the cold-induced effects on lipid peroxidation (*Figure 7C*). These results demonstrate that cold induces lipid peroxidation in the PVH and that our combined fiber photometry and BD-C11 approach can be reliably applied to evaluate the dynamics of brain lipid metabolism in a spatiotemporal manner.

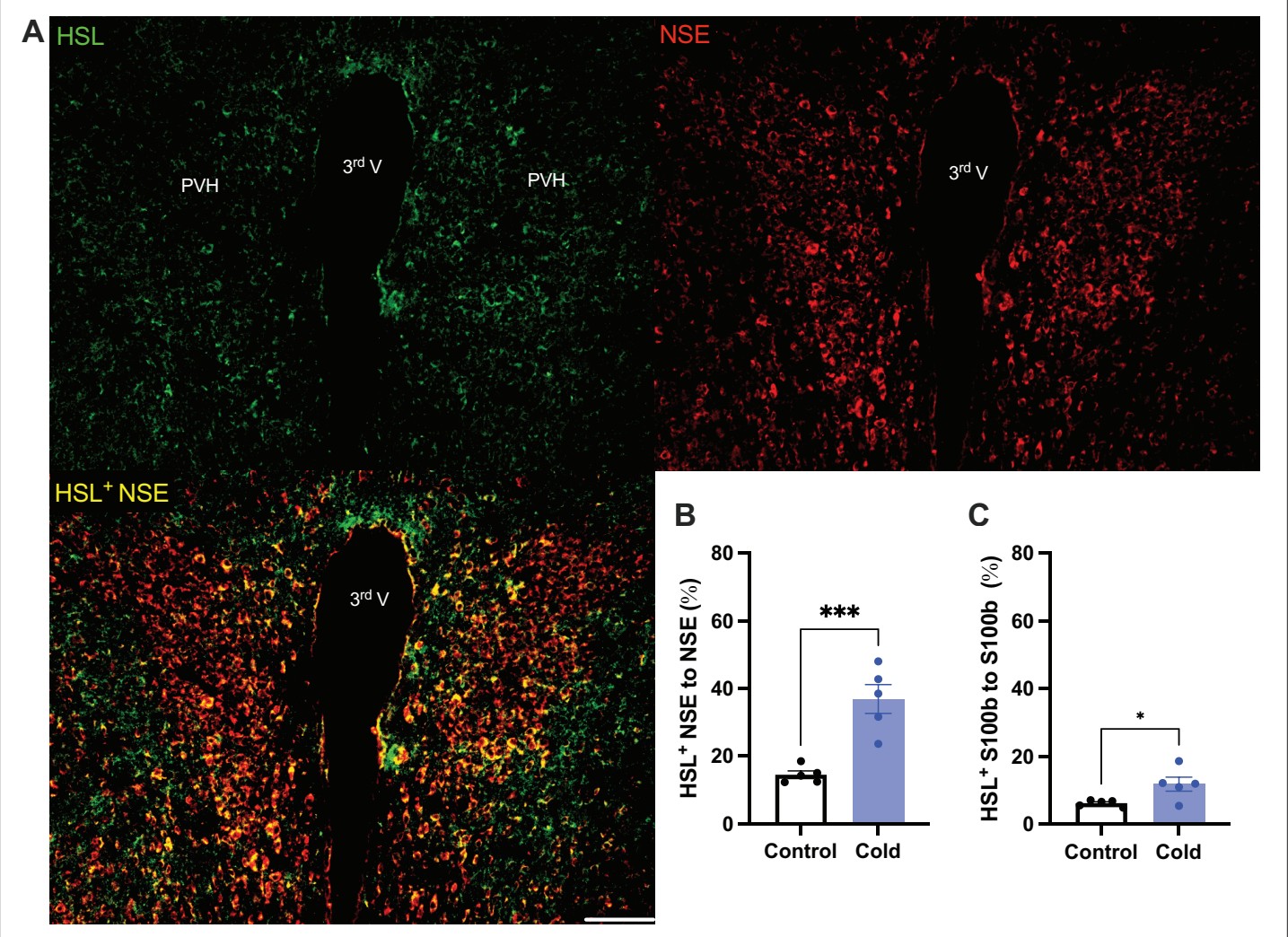

**Figure 3.** Cold increases HSL protein expressions in both neurons and astrocytes in paraventricular nucleus of the hypothalamus (PVH). Control or cold (4–6 hr)-challenged mouse brains were sectioned. HSL, NSE, and S100b were, respectively, co-stained using relevant antibodies. (**A**) Representative images of HSL (green), NSE (red), and HSL/NSE overlay (yellow) signals in PVH sections from cold-challenged mouse. (**B**) Group data of relative HSL/NSE overlay signals to total NSE signals in control and cold-challenged mice ($n = 5$ each group). (**C**) Group data of relative HSL/S100b overlay signals to total S100b signals in control and cold-challenged mice ($n = 5$ each group). Data represent mean ± SEM. Student $t$ tests were performed. ***$p < 0.001$, *$p < 0.05$. Scale bar, 100 μm for (**A**). PVH, paraventricular of hypothalamus; 3rd V, third ventricle.

The online version of this article includes the following source data for figure 3:

**Source data 1.** Group data of HSL expressing neurons and astrocyets.

## Lipid peroxidation is under the control of cell activity

We next sought to examine whether cell activity is required to modulate lipid peroxidation. To examine whether cold alters cell activity, we transduced PVH cells with the $Ca^{2+}$ indicator GCaMP$_{6f}$- and placed the PVH transduced and photometry fiber-implanted mice in a cold chamber (*Figure 7—figure supplement 1A*). Cold exposure increased the intensity of PVH GCaMP$_{6f}$ signals (*Figure 7—figure supplement 1B, C*), consistent with the increased Fos expressions (*Figure 6*). To inhibit cell activity, we utilized the GABA$_A$ receptor agonist muscimol (MUS) (*Sanders and Shekhar, 1995*; *Barbalho et al., 2009*) and glutamate receptor antagonist Kynurenic acid (KYN) (*Yoshida et al., 2012*). To verify the inhibitory effect of the combined two chemicals, we performed intra-PVH injection of both MUS and KYN in PVH GCaMP$_{6f}$-transduced and fiber-implanted mice. Thirty minutes post the MUS and KYN administration via intra-PVH injection was able to reduce cell activity, as indicated by the decreased intensity of GCaMP$_{6f}$ signals, compared to vehicle treatment (*Figure 7—figure supplement 1D, E*).

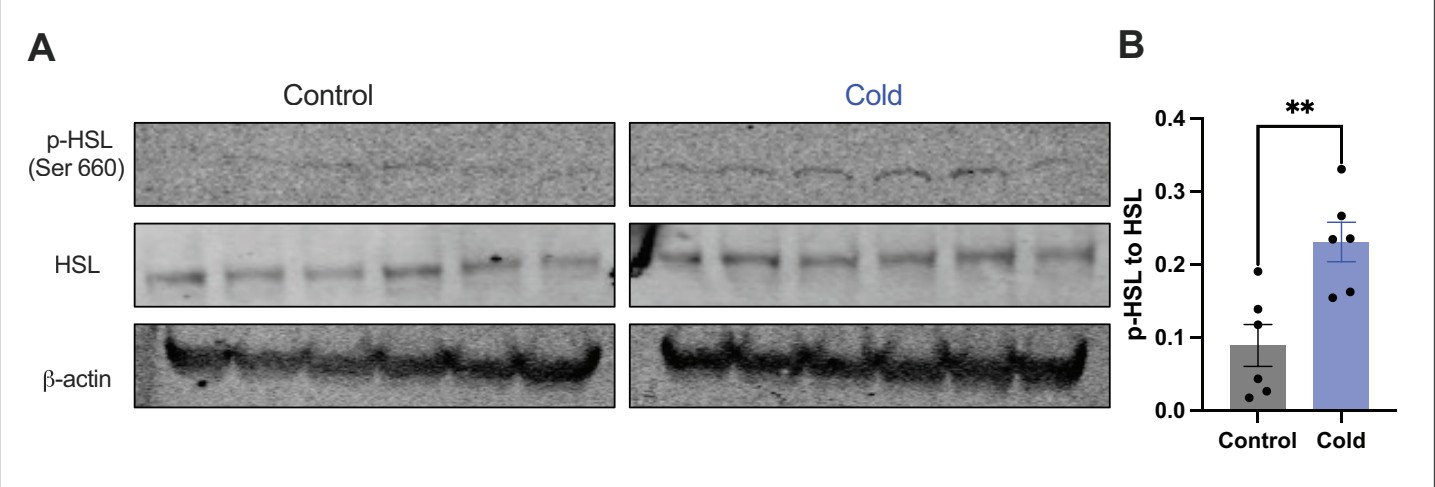

**Figure 4.** Cold increases the level of phosphorylated HSL in paraventricular nucleus of the hypothalamus (PVH). Micro-punches of PVH were collected in control and cold (4–6 hr)-challenged mice, which were used for (**A**) western blots of p-HSL (Ser660), HSL, and β-actin. (**B**) Group data of p-HSL fold change to HSL (n = 6 each group). Data represent mean ± SEM. Student *t* tests were performed. **p < 0.01. Each dot represents one mouse.

The online version of this article includes the following source data for figure 4:

**Source data 1.** Western blots of the HSL proteins in the PVH.

**Source data 2.** Gels of western blots of HSL, p-HSL, and actin.

**Source data 3.** Group data of western blots of HSL proteins in the PVH.

To test whether cell inactivation could diminish cold-induced effects on lipid peroxidation, intra-PVH injections of MUS and KYN were performed 30 min before placing the previously BD-C11-injected mice in a cold chamber and performing photometry monitoring of BD-C11 signals. Administration of the MUS and KYN prevented the cold-induced increase in the BD-C11 ratio (*Figure 7D*), suggesting that cell activation is required for the cold-induced lipid peroxidation.

## Cell inactivation prevents cold-induced lipolysis

To evaluate lipid lipolysis in the PVH, we performed intra-PVH injections of the EnzCheck lipase substrate through an implanted OmFC. The lipase substrate shifts its fluorescence to emission peak 515 nm (green) in the presence of lipases, and we detected the green signals using our fiber photometry system in a temporal manner. Intra-PVH injection of EnzCheck lipase substrate was performed 4 hr before placing the mice in a temperature-controlled chamber. Cold induced a potent increase in the green signals generated from the lipase substrate, indicating increased lipolysis (*Figure 8A, B*). We injected mice with the pan-lipase inhibitor diethylumbelliferyl phosphate (DEUP) via intra-PVT injections to inhibit lipolysis. Cold-induced lipolysis was reduced with the treatment of DEUP compared to vehicle-treated mice (*Figure 8C*). These results are consistent with the cold-induced increase in lipolytic markers (*Figures 1–4*), further indicating that cold induces lipid lipolysis in the PVH by increasing cytoplasmic lipases. To examine whether cell activity is required for the cold-induced lipolysis, in cohort groups of PVH-implanted mice which were treated with the lipase substrate via intra-PVH injections, we performed intra-PVH injections of the MUS and KYN 30 min before placing the mice in a cold chamber. Administration of MUS and KYN prevented the cold-induced conversion of the lipase substrate, indicating decreased lipolytic activity (*Figure 8D*). These findings suggest that cold induces a rapid cell activity-dependent lipid lipolytic effect.

## Cell inactivation reduces cold-induced LD accumulation

To verify our results of LDs detected in the fixed tissues (*Figure 5*), we next performed time-lapse photometry monitoring of dynamics of LDs in vivo in freely behaving mice. To achieve this goal, we implanted an OmFC cannula in PVH for LD marker BD493 (green) injections and monitoring in mice. Matching with the results collected from the fixed sections, cold induced an increase in the intensity of BD493 signals (*Figure 9A, B*). This result indicates that cold increases the formation and accumulation

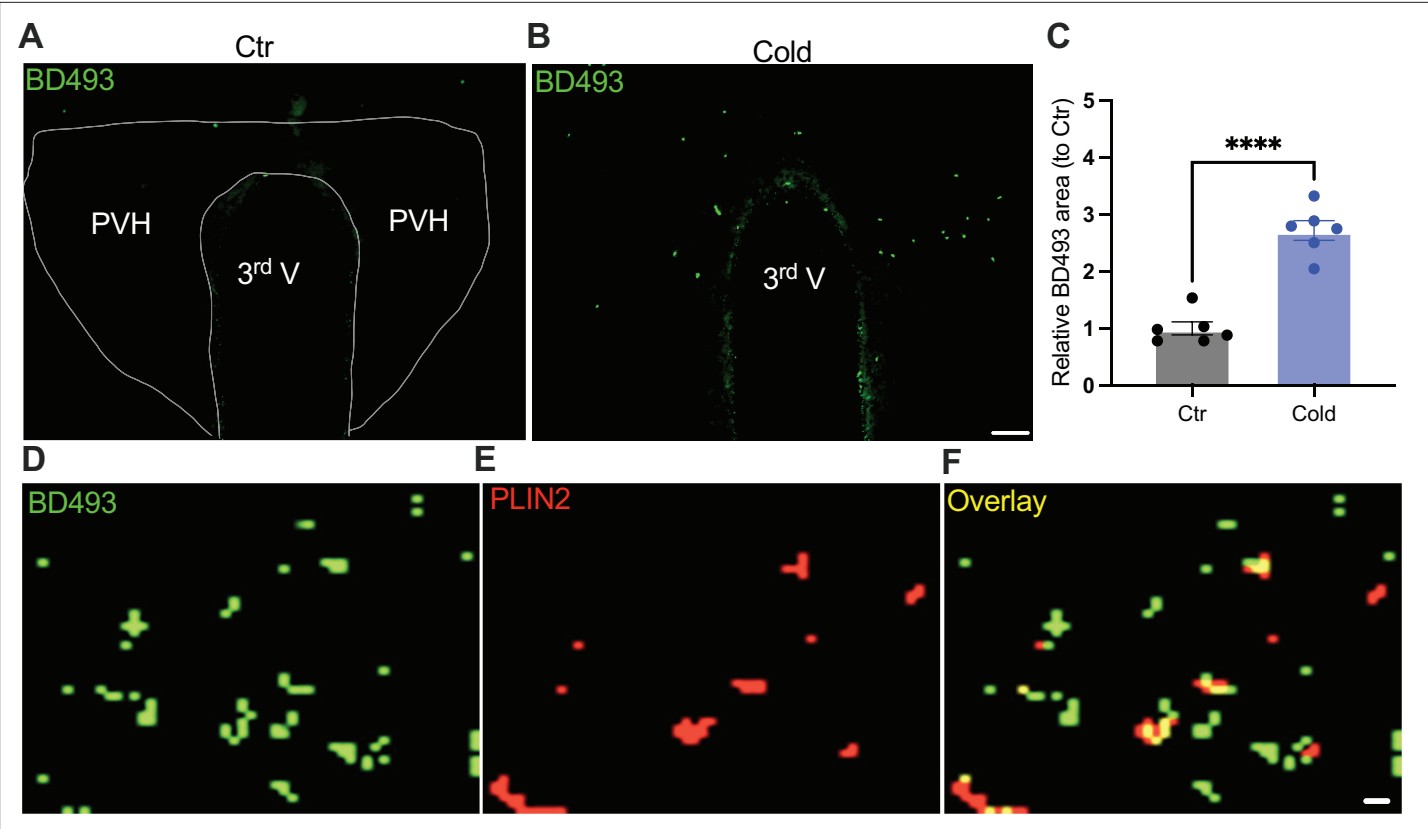

**Figure 5.** Cold-induced lipid droplet (LD) accumulation. Control or cold (30 min to 1 hr)-challenged mice were perfused and fixed. Mouse brains were sectioned. LDs in the paraventricular nucleus of the hypothalamus (PVH) were labeled using BODIPY493 (BD493). Representative images of BD493 signals in PVH sections from one control (**A**) and one cold-challenged mouse (**B**). (**C**) Group data of relative BD493-labeled area in PVH in control and cold-challenged mice (*n* = 6 each group). Sample images of BD493 (**D**) and perilipin2 (PLIN2) (**E**) and overlay (**F**) signals in the PVH. Data represent mean ± SEM. Student *t* tests were performed. ****p < 0.0001. Scale bars, 50 μm for (**A, B**) and 1 μm for (**D–F**). PVH, paraventricular of hypothalamus; 3rd V, third ventricle.

The online version of this article includes the following source data and figure supplement(s) for figure 5:

**Source data 1.** Group data of BD493 area in cold-challenged mice and control mice.

**Figure supplement 1.** Cold increases lipid droplet (LD) accumulation in astrocytes but not neurons.

**Figure supplement 1—source data 1.** Group data of BD493 areas in neurons and astrocytes.

**Figure supplement 2.** Group data of relative BD493-labeled area in paraventricular nucleus of the hypothalamus (PVH) in control or cold-challenged (4–6 hr) mice (*n* = 3 each group).

**Figure supplement 2—source data 1.** Relative LD area in the PVH in control mice and 4-6 hr cold challenged mice.

of LDs, which could be directly used for lipolysis by releasing fatty acids to mitochondria for β-oxidation. The lipid soluble antioxidant α-TP blocked the cold-induced LD accumulation (*Figure 9C*). To define a role of cell activity in modulating LDs, we performed intra-cannula injections of both MUS and KYN in BD493-injected mice in the PVH. MUS and KYN administration blunted the cold-induced accumulation of LDs, as cold failed to increase the BD493 signals in MUS- and KYN-treated mice compared to controls (*Figure 9D*).

## Discussion

For survival, it is crucial for individuals to precisely modulate lipid metabolism in both the peripheral tissues and brain in mammalian animals. Extensively studies have been focused on lipid metabolism in peripheral tissues particularly in adipose tissues, while little or nothing was known about that in the brain. This was probably in part because of limited techniques. By taking advantage of combined coordinated neurobiology methods and lipid metabolic assays, we provided in this study the in vivo

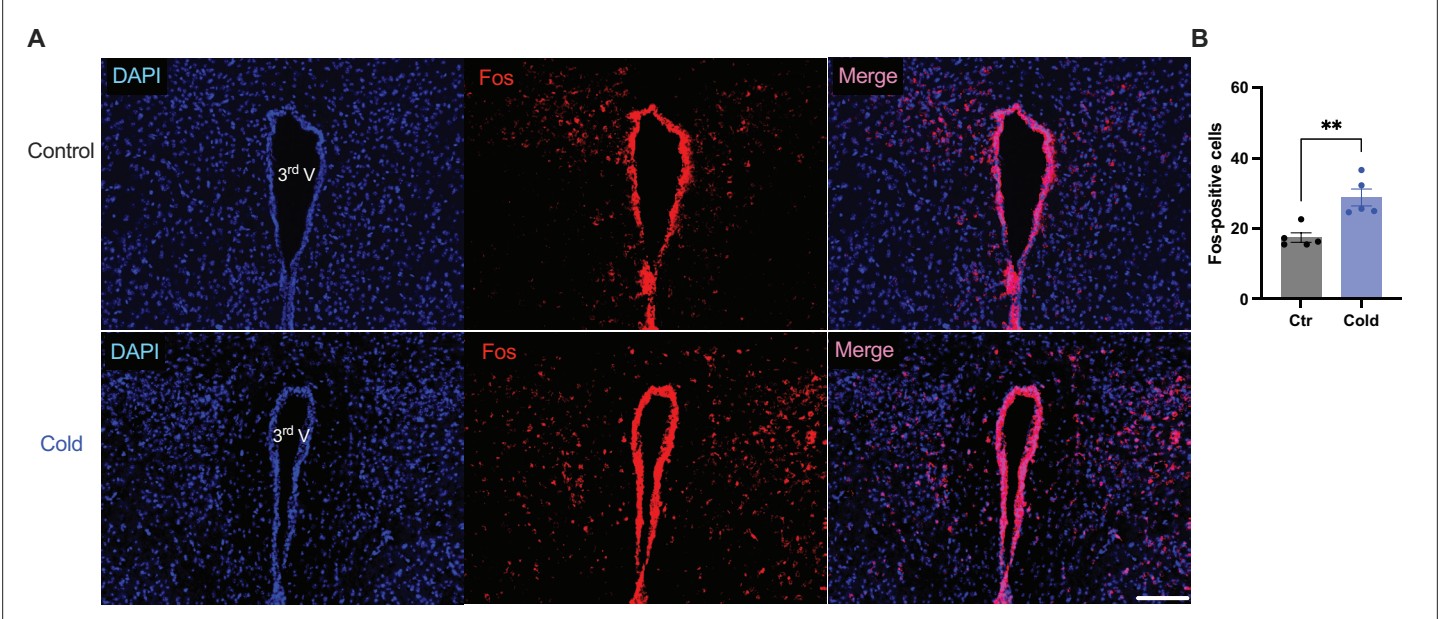

**Figure 6.** Cold increased Fos expressions in the paraventricular nucleus of the hypothalamus (PVH). Control or cold-challenged mouse brains were perfused and fixed and sectioned. Fos in the PVH were labeled using anti-Fos antibodies. (**A**) Representative images of Fos signals in PVH from one control (top) and one cold-challenged mouse (bottom). (**B**) Group data of Fos-positive cells in PVH in control and cold-challenged mice (*n* = 5 each group). Data represent mean ± SEM. Student *t* tests were performed. **p < 0.01. Scale bars, 100 μm for (**A**). 3rd V, third ventricle.

The online version of this article includes the following source data for figure 6:

**Source data 1.** Group data of Fos-positive cells in the PVH.

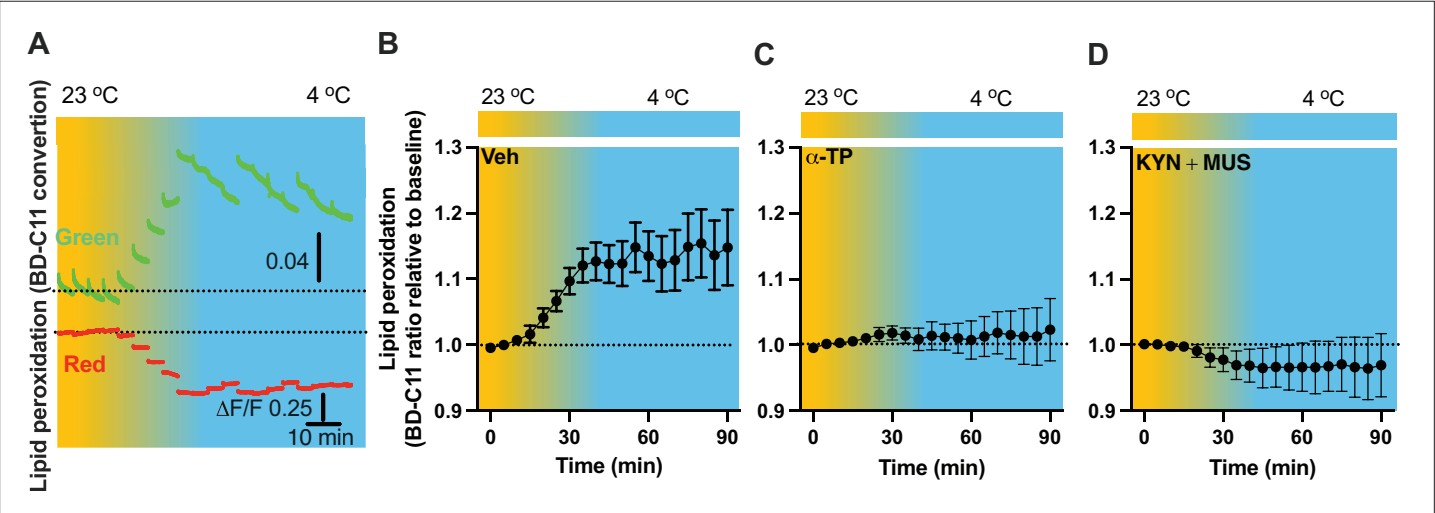

**Figure 7.** Cold-induced cell activity-dependent lipid peroxidation in the paraventricular nucleus of the hypothalamus (PVH). Mice were injected with the lipid peroxidation indicator BD-C11 in the PVH via the implanted optical fiber multiple fluid injection cannula (OmFC) 4 hr before placing them in a temperature-controlled chamber. Two-color fiber photometry was applied for time-lapse monitoring of BD-C11 conversion via the optic fiber of the OmFC. (**A**) Representative traces of real-time photometry monitoring of red and green signals simultaneously. Group data of relative BD-C11 ratio to baseline in mice treated with vehicle (**B**), *n* = 6, α-TP (**C**), *n* = 4, and KYN + MUS (**D**), *n* = 4. Data represent mean ± SEM.

The online version of this article includes the following source data and figure supplement(s) for figure 7:

**Source data 1.** Group data of photometry monitoring of lipid peroxidation in the PVH.

**Figure supplement 1.** Neuronal inhibition by KYN + MUS.

**Figure supplement 1—source data 1.** Group data of photometry monitoring of PVH neuronal activity.

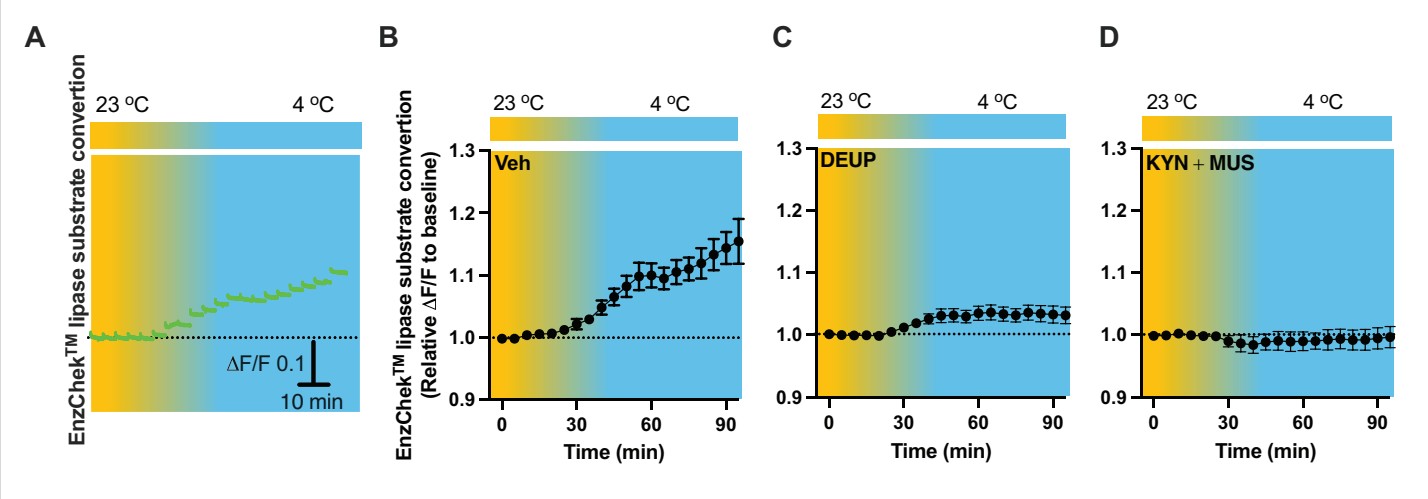

**Figure 8.** Cold-induced cell activity-dependent lipid lipolysis. Mice were injected with the lipase substrate in the paraventricular nucleus of the hypothalamus (PVH) via the implanted optical fiber multiple fluid injection cannula (OmFC) 4 hr before placing them in a temperature-controlled chamber. Two-color fiber photometry was used for time-lapse monitoring of lipase substrate conversion via the optic fiber of the OmFC. (**A**) Representative traces of real-time photometry monitoring of green signals. Group data of lipase substrate conversion in mice treated with vehicle (**B**), $n = 4$, $\alpha$-TP (**C**), $n = 4$, and KYN + MUS (**D**), $n = 4$. Data represent mean ± SEM.

The online version of this article includes the following source data for figure 8:

**Source data 1.** Group data of photometry monitoring of lipid lipolysis in the PVH.

evidence that cold is capable to induce acute cell activity-dependent central nervous system lipid metabolic activities, including lipid peroxidation and mobilization, LD formation, and lipid lipolysis in the hypothalamus. These results fill in a missing but important gap in our understanding of brain region-selective lipid metabolism in physiological conditions, such as cold applied in this study. Also, our in vivo time-lapse fiber photometry approach detecting probe-labeled lipid metabolites would provide a powerful alternative method or technique to evaluate the dynamics of lipid metabolism in

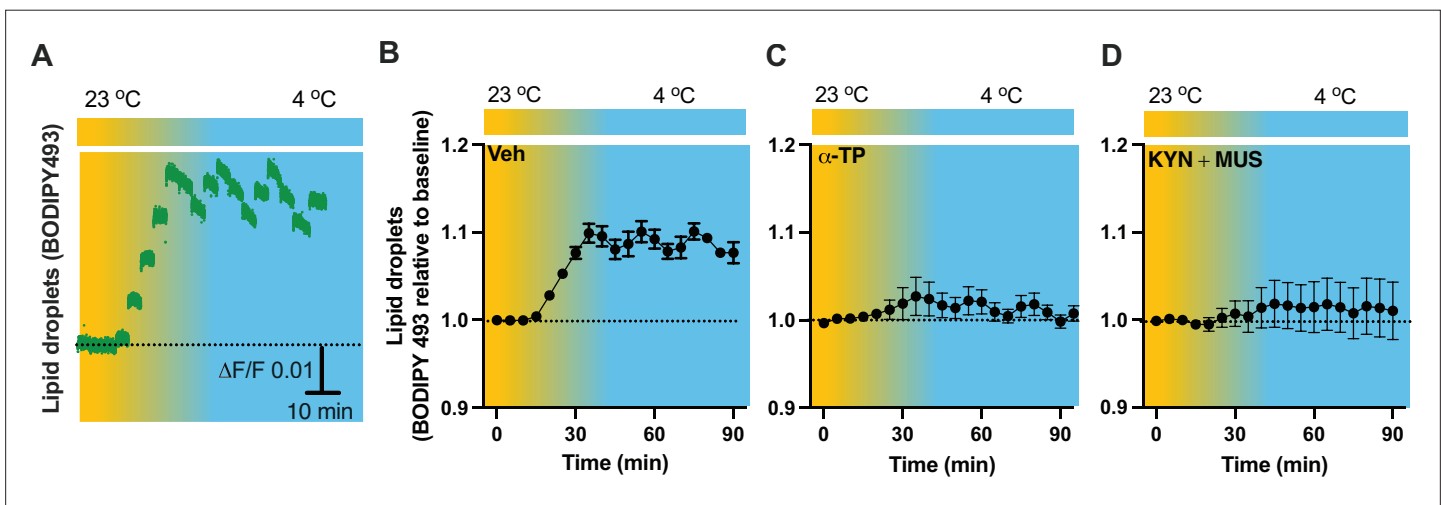

**Figure 9.** Cold-induced cell activity-dependent lipid droplet (LD) accumulation. Mice were injected with the LD marker BODIPY493 in the paraventricular nucleus of the hypothalamus (PVH) via the optical fiber multiple fluid injection cannula (OmFC) 4 hr before placing them in a temperature-controlled chamber. Fiber photometry was used for time-lapse monitoring of BODIPY493 signals via the optic fiber of the OmFC. (**A**) Representative traces of real-time photometry monitoring of BODIPY493 signals. Group data of BODIPY493 in mice treated with vehicle (**B**), $n = 4$, $\alpha$-TP (**C**), $n = 4$, and KYN + MUS (**D**), $n = 6$. Data represent mean ± SEM.

The online version of this article includes the following source data for figure 9:

**Source data 1.** Group data of photometry monitoring of LD dynamics in the PVH.

live animals. We believe that our studies will also be paradigmatic for understanding abnormal lipid metabolism in brain regions that modify energy metabolism in metabolic disorders such as obesity and diabetes.

The third ventricle adjacent PVH is a conserved brain region (*Machluf et al., 2011*), composed of different cell populations including those multiple brain regions-projecting parvocellular neurons and magnocellular neurons. Particularly, PVH plays varied crucial roles in modulating the hypothalamic–pituitary–adrenal (HPA) axis and the hypothalamic–pituitary–thyroid (HPT) axis (*Swanson and Sawchenko, 1983*). This determines a key role of the PVH in modulating body functions, energy metabolism and glucose homeostasis in health and disease. Our findings in this study show that cold increases the expressions levels of Fos, lipid peroxidation, LD accumulation, and lipolytic markers in the PVH. These findings reveal a new role for the PVH in preventing fatty acid toxicity in pathological conditions and in integrating cold-induced broader regulatory responses.

Activated neurons would induce phospholipid peroxidation to release toxic fatty acids. One recent study (*Ioannou et al., 2019*) shows that nearby astrocytes have the capacity to endocytose neuron-released fatty acids, store fatty acids in LDs, and liberate fatty acids from LDs through lipolytic processes to mitochondria for β-oxidation. Consistently, our data in this study demonstrate that cold-induced lipid peroxidation and lipolysis in the brain are also under the control of cell activities. Our data show that cold increases expressions of ATGL and HSL in both neurons and astrocytes (*Figures 2 and 3*). These findings suggest that neurons and astrocytes communicate and interact each other to modulate lipid metabolism and prevent fatty acid accumulations in the brain in certain physiological and pathological conditions.

Lastly, the brain is highly thermal sensitive (*Brooks, 1983*). Identifying brain regions sensitive to ambient temperature is significant and important in our understandings of brain energy metabolism and homeostasis, as which might provide potential targetable sites in the treatment of brain lipid metabolism-associated neurological and psychological disorders.

## Materials and methods

The experimental protocol (#00001306) was approved by the Institutional Animal Care and Use Committees at the Albert Einstein College of Medicine and conducted following the U.S. National Institutes of Health guidelines for animal research.

### Animals

C57/BL6J wild-type mice (#000664, Jackson Laboratory) have been described previously and are available from The Jackson Laboratory. Both male and female mice (age 8–12 weeks) were used at the start of experiments. Mice were group-housed 3–5 mice per cage in humidity- and temperature (22–25°C)-controlled rooms on a 12-hr light:dark cycle (lights on from 8:00 a.m. to 8:00 p.m.) with ad libitum access to water and mouse regular chow (#5001, LabDiet). Mice were single-caged after they received viral transductions with or without guide or optic fiber cannula insertion until all the experimental procedures were finished.

### Pharmacology

All the chemicals were purchased from Sigma except where noted. For the experiments requiring intra-PVH injections, an injector with 1 mm extension beyond the custom-made OmFC (Doric Lens) implanted over the PVH was attached by polyethylene tubing to a Hamilton syringe. The injection was performed at a speed of 50 nl per min for 4 min using a matched fluid injector consisting of a 1.25-mm ferrule and a sleeve connector, and the injector was withdrawn 10 min after the final injection. Grip cement (DENTSPLY) was used to anchor the cannula to the skull, and a plug was inserted to keep the cannula from becoming clogged when the injector was not in place. Mice were then returned to the home cage for 1 week at least before the experiments. The amount for intra-PVH injection was 200 nl of vehicle or chemicals (in μM): 100 BODIPY 493/503 (D3922, Invitrogen), 100 BODIPY 581/591 C11 (D3861, Invitrogen), 1 MitoSOX Mitochondrial Superoxide Indicators (M36005, Invitrogen), 1 EnzChek Lipase Substrate 505/515 (E33955, Invitrogen), 100 DEUP (D7692, Sigma), and 500 α-Tocopherol (#258024, Sigma); or 200 nl of 250 pmol Muscimol (M1523, Sigma) and 100 mol Kynurenic acid (K3375, Sigma).

## Stereotaxic OmFC implantations for intra-PVH injections and photometry

Following our previously documented protocols (*Chen et al., 2022*), mice were anesthetized with 3% isoflurane to induce the anesthesia and with 1.5–2.0% isoflurane to maintain anesthesia during the surgery and placed in a stereotaxic frame (Kopf Instruments). A small incision was made in the skin of the head, a small hole was drilled on the skull, and mice were implanted with a customer-made OmFC (Doric Lens) over the PVH (coordinates from bregma: AP –0.8 mm, 0.2 mm from midline, DV –4.0 mm). The cannula was fixed to the skull with stainless steel screws and dental cement. After the surgery, all mice received meloxicam (5 mg/kg) and continued to be housed individually. Two weeks after surgery, the mice were briefly anesthetized and inserted with the fluid injector (with 1 mm protrusion) into PVH. For intra-PVH injection, the injector was attached to a Hamilton syringe through a polyethylene tube and the mice received 200 nl of vehicle, chemicals as listed in the Pharmacology section, or viral vectors (AAV$_5$-CamKII-GCaMP$_{6f}$-WPRE-SV40, addgene#100834, titer, $2.3 \times 10^{13}$ GC/ml), at a rate of 50 nl/min.

## Micro-punches of hypothalamic nuclei

As described in our previously published studies (*Qi and Yang, 2015*; *Zhang et al., 2020*; *Chen et al., 2022*), acute brain slices that include hypothalamic PVH, LH, DMH, VMH, and ARC, respectively, were prepared. Briefly, mice were first placed in a cold chamber (4°C) for 30 min or 4–6 hr, respectively, as described in the text and figure legends. After the cold exposure, mice were deeply anesthetized with isoflurane and decapitated. The mouse brains were dissected rapidly and placed in ice-cold oxygenated (95% $O_2$ and 5% $CO_2$) solution containing the following (in mM): 110 choline chloride, 26.2 D-glucose, 2.5 KCl, 1.25 NaH$_2$PO$_4$, 2 CaCl$_2$, 7 MgSO$_4$, 11.6 Na-L-ascorbic acid, and 3.1 Na-pyruvate. Coronal brain slices (260 µm thick) were cut with a vibratome (Leica; VT 1200 S) and maintained in an incubation chamber containing ice-cold ACSFs (in mM): 119 NaCl, 25 NaHCO$_3$, 11 D-glucose, 2.5 KCl, 2.5 CaCl$_2$, 1.3 MgSO$_4$, and 1 NaH$_2$PO$_4$. Micro-punches of the PVH, LH, DMH, VMH, and ARC were performed using a punch (1.5 mm diameter; Stoelting#57403) or a pipette tip (1.5 mm diameter). Six micro-punches of each region were obtained from each mouse were, respectively, collected and immediately placed in 200 µl Trizol reagent (Invitrogen) and stored at –80°C until analysis.

## Total RNA extractions and real-time qPCR

Total RNA was extracted using Trizol reagent (Invitrogen) according to the manufacturer's instruction. Briefly, the collected tissues were lysed in 200 µl of Trizol reagent (Invitrogen) and 40 µl of chloroform was added into each tube and subsequently vortexed for 15 s. After incubation for 2 min, centrifuged at $12,000 \times g$ for 10 min (4°C). The aqueous phase was transferred to fresh tube and equal volume of isopropanol was added. The tube was incubated for 20 min and centrifuged at $12,000 \times g$ for 10 min (4°C). The RNA pellet was washed with 70% ethanol by vortex and centrifuged at $12,000 \times g$ for 10 min (4°C). cDNA was synthesized by High-Capacity cDNA Reverse Transcription Kit (Applied Biosystems) following the manufacturer's instructions. Real-time qPCR was performed using QuantStudio 3 instruments (Applied Biosystems). The mixture for qPCR reaction was prepared in a final volume of 20 µl containing 1 µl cDNAs and 10 µl of LightCycler 480 SYBR Green I Master (Roche) in the presence of primers at 500 nM. The specific primer sequences used were the following:

The specific primer sequences used were the following:

*Pnpla2* forward, 5′-CCAACACCAGCATCCAGT-3′;
*Pnpla2* reverse, 5′-CAGCGGCAGAGTATAGGG-3′;
*Lipe* forward, 5′-CGCCATAGACCCAGAGTT-3′
*Lipe* reverse, 5′-TCCCGTAGGTCATAGGAGAT-3′;
*Cidea* forward, 5′-TGCTCTTCTGTATCGCCCAGT-3′
*Cidea* reverse, 5′-GCCGTGTTAAGGAATCTGCTG-3′;
*Prdm16* forward, 5′-CCACCAGCGAGGACTTCAC-3′
*Prdm16* reverse, 5′-GGAGGACTCTCGTAGCTCGAA-3′;
*Ucp2* forward, 5′-CAGAGCACTGTCGAAGCCTA-3′
*Ucp2* reverse, 5′-GTATCTTTGATGAGGTCATA-3′;
*Actb* forward, 5′-GCTGTCCCTGTATGCCTCT-3′

*Actb* reverse, 5′-GTCTTTACGGATGTCAACG-3′;.

The relative level of expression was calculated using the comparative $2^{-\Delta\Delta CT}$ method.

## Two-color two-channel fiber photometry

Briefly, three excitation wavelengths were used: 560, 505/490, and 405 nm. Excitation lights were generated through fiber-coupled two connectorized LEDs (CLED_560 for 560 nm; CLED_505/490 for 505 or 490 nm; CLE_405 for 405 nm; Doric Lenses) driven by a four-channel LED driver (LEDD_4; Doric Lenses). The LEDD_4 was controlled by a fiber photometry console (Doric Lenses) connected to a computer. Excitation lights were passed through two fluorescence MiniCubes (iFMC6_IE(400-410)_ E1(460-490)_F1(500-540)_E2(555-570)_F2(580-680)_S)-6 ports with two integrated photodetector heads. The single detector measures both signals within the fluorescence detection windows from 500 to 540 nm and 580 to 680 nm band. The combined excitation light was sent into a patch cord made of a 400-µm core, 0.48 NA, low-autofluorescence optical fiber (Doric Lenses). The patch cord was connected to the implanted OmFC consisting of a 1.25-mm diameter optic fiber via a sleeve (Doric Lenses; Zirconia Sleeve 1.25 mm with black cover; Sleeve_ZR_1.25-BK). Both green and red fluorescence signals were collected through the same patch cord and passed through the same Minicube and focused onto a Fluorescence Detector Head (FDH; Doric Lenses). The photometry experiments were run in a Lock-in mode, and the acquisition rate was set to 12.0 ksps*C controlled by Doric Neuroscience Studio software (Doric Lenses). The fiber photometry experiments were performed 5 min after connecting the optic fibers to the animals. We processed the signals using the Doric Neuroscience Studio software (V5.3.3.14) to calculate the normalized fluorescence variation of the images ($\Delta F/F$), and averaged the signals at 1 s bins. To avoid or minimize bleaching over time, we performed patch cord photobleaching for 12 hr before each experiment, reduced the illumination power outputs as much as possible, and recorded the signals for 30 s every 5 min. We used a rotary joint for long-term photometry recordings in freely moving animals.

## Brain histology and immunofluorescence

Mice were euthanized and transcardially perfused with 1× phosphate-buffered saline (PBS, pH 7.4) followed by 4% paraformaldehyde in phosphate buffer (PFA, pH 7.2). Mouse brains were removed and post-fixed in 4% PFA overnight. Fixed brains were transferred to 30% sucrose in PFA for cryoprotection for three days. Next, 30 µm coronal sections were cut in a freezing cryostat (Leica).

For immunofluorescence, slices were washed three times in 1× PBS for 10 min each and heat incubated with target antigen retrieval solution (Invitrogen) for 2 min at 95°C. And then, the slices were washed three times in 1× PBS for 10 min each, followed by permeabilization in 1% Triton X-100 solution in 1× PBS for 40 min at room temperature. After blocking for 1 hr at room temperature, the slices were incubated with a mouse Fos antibody (1:150, sc-166940 AF647, Santa Cruz Biotechnologies), rabbit ATGL (1:150, bs-3831R-BF488, Bioss), rabbit HSL (1:150, bs-3223R-BF488, Bioss), rabbit NSE (1:200, bs-10445R-BF594, Bioss), and mouse S100b (1:200, bsm-10832-BF594, Bioss), respectively, for overnight at 4°C in the dark. The slices were then rinsed three times in 0.1% Triton X-100 solution in 1× PBS for 10 min each. For LD staining, slices were washed three times in 1× PBS for 10 min each and then incubated with 1% Triton X-100 solution in 1× PBS for 40 min. After blocking for 1 hr, the slices were stained with 20 µg/ml BODIPY 493/503 (Invitrogen, D3922) for 4 hr at room temperature in the dark. After the incubation, the slices were rinsed three times in 0.1% Triton X-100 solution in 1× PBS for 10 min each.

For BODIPY 493/503 co-staining with perilipin-2, NSE, and S100b, respectively, after permeabilization in 1% Triton X-100 solution in 1× PBS for 40 min and blocking for 1 hr at room temperature, the slices were incubated with rabbit perilipin-2 (1:200, CL594-15294, Proteintech), rabbit NSE, and mouse S100b antibodies, respectively, for overnight at 4°C in the dark. The slices were then washed three times in 0.1% Triton X-100 solution in 1× PBS for 10 min and incubated with 20 µg/ml BODIPY 493/503 (Invitrogen, D3922) for 4 hr at room temperature in the dark and then washed three times in 0.1% Triton X-100 solution in 1× PBS for 10 min each time. The stained slices were dried and mounted with mounting medium (0100-20, Southern Biotech). Images were taken using the All-in-One Fluorescence Microscope (BZ-X800E Keyence) and analyzed using the BZ-X800 Analyzer (Keyence).

## PVH tissue western blot

Acute coronal brain slices (260 μm thickness) were collected following the procedures as stated in the above. Six micro-punches of PVH sections from each mouse were made and immediately placed in 40 μl cold lysis buffer containing 50 mM Tris–HCl, 150 mM NaCl, 0.25% Na-deoxycholate, 1 mM EDTA, 1% NP-40, 1 mM PMSF (20 mM PMSF in 100% ethanol), and 1 protease inhibitor tablet (A32955, Thermo Scientific). The samples were homogenized in lysis buffer and centrifuged at 14,000 × rpm at 4°C for 20 min. The supernatants were transferred to a fresh tube and used for Bradford assay to quantify the protein concentration. Equal amounts of protein from each sample were denatured and boiled at 100°C for 5 min. Proteins were separated by electrophoresis in 10% SDS–PAGE gel and transferred to PVDF membrane. The membrane was blocked with Protein-Free blocking buffer (927-80001, LI-COR) for 1 hr and incubated overnight with relevant primary antibodies at 4°C, including the p-HSL (1:1000, PA5-64494, Invitrogen), HSL (1:1000, PA5-17196, Invitrogen), and β-actin (1:5000, MAB8929, R&D Systems). After washing in PBS-Tween 20 (PBST), the membranes were incubated with secondary antibodies for 1 hr and a subsequent PBST washing. Images were acquired using Odyssey Classic Imager (LI-COR). The band intensity of HSL and p-HSL was quantified using the ImageJ.

## Statistics and reproducibility

Animals were randomly assigned to experimental or control groups. Following post hoc histological conformation of viral transfections and cannula/fiber placements, all mice with inaccurate viral infections or cannula placements were excluded from behavior experiments. All experiments were repeated at least twice, and data represent the technical and biological replicates. Paired or unpaired Student's $t$ tests were used to analyze differences between two groups of the same or different mice when appropriate, respectively. One-way ANOVA with Tukey's post hoc test was used to compare group data from more than two groups of mice. Two-way ANOVA was used to analyze data from more than two groups across various time points. All data were analyzed by using Prism 10.0 (GraphPad Software).

# Acknowledgements

We thank all the members of the Yang laboratory for discussion and critical comments on this study. We thank the Doric Lenses for helping us build and adjust the two-channel two-color photometry rig. For the genetically encoded calcium indicator GCaMP$_{6f}$ plasmids, we thank Dr. James M Wilson for depositing the plasmid of pENN-AAV-CamKII-GCaMP$_{6f}$-WPRE-SV40. This work was supported by the NIH (R01 DK112759, R01 DK135717 to Y.Y.) and Einstein Research Foundation.

# Additional information

### Competing interests

Yunlei Yang: Reviewing editor, *eLife*. The other authors declare that no competing interests exist.

### Funding

| Funder | Grant reference number | Author |
| --- | --- | --- |
| National Institutes of Health | DK112759 | Yunlei Yang |
| National Institutes of Health | DK135717 | Yunlei Yang |

The funders had no role in study design, data collection and interpretation, or the decision to submit the work for publication.

### Author contributions

Hyeonyoung Min, Data curation, Formal analysis, Investigation, Methodology, Writing - original draft, Writing - review and editing; Yale Y Yang, Writing - review and editing; Yunlei Yang, Conceptualization,

Data curation, Formal analysis, Supervision, Funding acquisition, Validation, Investigation, Visualization, Methodology, Writing - original draft, Project administration, Writing - review and editing

### Author ORCIDs
Yunlei Yang  https://orcid.org/0000-0002-7623-6680

### Ethics
Experimental protocols were approved by the Institutional Animal Care and Use Committees at the Albert Einstein College and conducted following the U.S. National Institutes of Health guidelines for animal research. The experimental protocol (#00001306).

Reviewer #1 (Public review): https://doi.org/10.7554/eLife.98353.3.sa1
Author response https://doi.org/10.7554/eLife.98353.3.sa2

## Additional files

### Supplementary files
MDAR checklist

### Data availability
All data were presented in the text figures, figure supplements, and source data.

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
