## [Editor Report · eLife Assessment]

This phenomenological study reported that cold exposure induced mRNA expression of genes related to lipid metabolism in the paraventricular nucleus of the hypothalamus (PVH). While the paper does not address cell-type specificity or the functional role of lipids in PVH, the findings might still serve as a **useful** basis for others to explore their relevance to brain responses to cold. In the revised manuscript, the authors made adequate editions, such as new immunostaining and immunoblotting of AGTL and HSL in the PVH, and pharmacological inhibition of lipid peroxidation and lipolysis. The authors also increased the sample size of some experiments and revised the text to limit their data interpretation. Thus, the reviewers considered that these studies are **solid** in conclusively describing how the PVH is reprogrammed at the level of gene expression by cold exposure.

---

## [Referee Report · Reviewer #1 (Public review)]

Summary:

This study focuses on metabolic changes in the paraventricular hypothalamic (PVH) region of the brain during acute periods of cold exposure. The authors point out that in comparison to the extensive literature on the effects of cold exposure in peripheral tissues, we know relatively little about its effects on the brain. They specifically focus on the hypothalamus, and identify the PVH as having changes in Atgl and Hsl gene expression changes during cold exposure. They then go on to show accumulation of lipid droplets, increased Fos expression, and increased lipid peroxidation during cold exposure. Further, they show that neuronal activation is required for the formation of lipid droplets and lipid peroxidation.

Strengths:

A strength of the study is trying to better understand how metabolism in the brain is a dynamic process, much like how it has been viewed in other organs. The authors also use a creative approach to measuring in vivo lipid peroxidation via delivery of BD-C11 sensor through a cannula to the region in conjunction with fiber photometry to measure fluorescence changes deep in the brain.

Comments on revised version:

The authors have attempted to address concerns brought to their attention in the initial review. They have performed one or two additional experiments to address concerns (e.g. adding fiber photometry of PVH neurons and trying to manipulate lipid peroxidation) though many of the concerns from the original review stand. The authors have also revised the text to limit the extent of their claims and to improve clarity, which is appreciated.

---

## [Author Response]

The following is the authors’ response to the original reviews.

We were pleased that many of the critical comments of the reviewers have allowed us to improve our manuscript. In addition to revise the originally submitted figures, we performed new experiments (e.g. new Fig.2, Fig.3, Fig.4, and Fig.6) and revised the manuscript substantially following the reviewers’ comments and suggestions to our initial submission. A point-by-point response to the reviewers’ critiques are summarized below, and new supportive data are provided in this revised manuscript. Per the Reviewers’ comments and revisions, we revised the title to be “Cold induces brain region-selective cell activity-dependent lipid metabolism”.

**Reviewer #1**:Strengths:A strength of the study is trying to better understand how metabolism in the brain is a dynamic process, much like how it has been viewed in other organs. The authors also use a creative approach to measuring in vivo lipid peroxidation via delivery of a BD-C11 sensor through a cannula to the region in conjunction with fiber photometry to measure fluorescence changes deep in the brain.

We thank the Reviewer so much for the positive comments on this interesting study on metabolism in the brain.

Weaknesses:One weakness was many of the experiments were done in a manner that could not distinguish between the contributions of neurons and glial cells, limiting the extent of conclusions that could be made. While this is not easily doable for all experiments, it can be done for some. For example, the Fos experiments in Figure 3 would be more conclusive if done with the labeling of neuronal nuclei with NeuN, as glial cells can also express Fos. To similarly show more conclusively that neurons are being activated during cold exposure, the calcium imaging experiments in Figure S3 can be done with cold exposure.

We agreed with the Reviewers’ comments. We revised the original Figure 3 (new Figure 6) and Figure S3 (new Figure S4). Our data show that cold increased Fos-positive cells in the PVH (Figure 6) and increased neuronal Ca2+ signals (new Figure S4). As it is difficult to exclude the involvements of astrocytes in the cold-induced lipid metabolism, and to address this reviewer’s questions, we revised the title and the text with replacing “neuronal” with “‘cell” activity, and we concluded that cold induced lipid metabolism depending on “cell activity” instead of “neuronal activity”. Studying cell type-specific contributions to the cold-induced effects on lipid metabolism will require many efforts beyond the scope of this study, to which we assumed that both neurons and glial cells contribute.

Additionally, many experiments are only done with the minimal three animals required for statistics and could be more robust with additional animals included.

We thank this reviewer for the comments. We added the sample sizes accordingly in this revised manuscript.

Another weakness is that the authors do not address whether manipulating lipid droplet accumulation or lipid peroxidation has any effect on PVH function (e.g. does it change neuronal activity in the region?).

We thank this reviewer for bringing up this interesting point. The focus of this study was to examine how cold modulates lipid metabolism in the brain, while it is another interesting project studying how brain lipid metabolism (e.g. manipulating LD accumulation or lipid peroxidation) modulates neuronal activity, which however will require many efforts beyond the scope of this study. Manipulating LD or peroxidation would affect multiple cellular signaling pathways and physiological experimental conditions need to be developed. However, to address this reviewer’s questions, we performed preliminary studies with treating brain slices with the lipid peroxidation inhibitor a-TP and recorded PVH neurons, but did not observe differences in firing rates in a-TPtreated brain slices and controls (Data not shown).

**Reviewer #2**:Strengths:A set of relatively novel and interesting observations. Creative use of several in vivo sensors and techniques.

We thank the Reviewer so much for the positive comments on our studies in both concept and techniques.

Weaknesses:(1) The physiological relevance of lipolysis and thermogenesis genes in the PVH. The authors need to provide quantitative and substantial characterizations of lipid metabolism in the brain beyond a panel of qPCRs, especially considering these genes are likely expressed at very low levels. mRNA and protein level quantification of genes in Fig 1, in direct comparison to BAT/iWAT, should be provided. Besides bulk mRNA/protein, IHC/ISH-based characterization should be added to confirm to cellular expression of these genes.

We agreed with the Reviewer’s comments and thank this reviewer for the constructive suggestions. To address this reviewer’s comments and suggestions, we performed additional experiments to verify cold-induced expressions of lipid lipolytic genes and proteins. For example, we stained ATGL and HSL in both neurons and astrocytes in the PVH. Matching with the increased gene expressions, cold increased protein expressions of ATGL (new Figure 2) and HSL (new Figure 3) in both neurons and astrocytes. We also performed western blots of p-HSL and HSL and observed that cold increased the expression level of p-HSL (new Figure 4). These new results support our conclusions and further demonstrate that cold increases lipid metabolism in the PVH.

(2) The fiberphotometry work they cited (Chen 2022, Andersen 2023, Sun 2018) used well-established, genetically encoded neuropeptide sensors (e.g., GRABs). The authors need to first quantitatively demonstrate that adapting BD-C11 and EnzCheck for in vivo brain FP could effectively and accurately report peroxidation and lipolysis. For example, the sensitivity, dynamic range, and off-time should all be calibrated with mass spectrometry measurements before any conclusions can be made based on plots in Figures 4, 5, and 6. This is particularly important because the main hypothesis heavily relies on this unvalidated technique.

We thank this reviewer’s comments. Fiber photometry has been well demonstrated to detect fluorescent-labelled biomolecules in my laboratory and other labs, as indicated in the above stated publications. In this study, we combined photometry with the well commercially developed and validated lipid metabolic fluorescent-labelled biomarkers to monitor lipid metabolic dynamics in vivo. We indeed verified this approach in both brain (this study) and peripheral adipose tissues (another project). Particularly, our data in this study show that lipid peroxidation inhibitor a-TP blocked the cold-induced lipid peroxidation signals (Fig. 7A-C) and the pan-lipase inhibitor DEUP blocked the cold-induced lipolytic signals (Fig. 8A-C). These results demonstrate that the signals detected by photometry indeed reflect lipid peroxidation and lipolysis respectively in the brain. Meanwhile, we agreed with the reviewer’s suggestions on mass spectrometry measurements, while it is not feasible for us to perform the spectrometry in the brain in vivo at this moment.

(3) Generally, the histology data need significant improvement. It was not convincing, for example, in Figure 3, how the Fos+ neurons can be quantified based on the poor IF images where most red signals were not in the neurons.

We thank this reviewer for this comment. We performed additional experiments to add sample size and presented high quality images.

(4) The hypothesis regarding the direct role of brain temperature in cold-induced lipid metabolism is puzzling. From the introduction and discussion, the authors seem to suggest that there are direct brain temperature changes in responses to cold, which could be quite striking. However, this was not supported by any data or experiments. The authors should consolidate their ideas and update a coherent hypothesis based on the actual data presented in the manuscript.

We thank this reviewer for bringing up this comment and constructive suggestions. To make this study more concise on the cold-induced lipid metabolism, we removed the statements related to the brain temperature.

**Reviewer #1 (Recommendations For The Authors):**
An additional minor weakness is that the authors are redundant in their discussion, sometimes repeating sections from the introduction (e.g. this line in the discussion "Evidence shows that the brain's energy expenditure efficiency largely depends on the temperature (Yu et al., 2012), and temperature gradients between different brain regions exist (Anderson and Moser, 1995; Delgado and Hanai, 1966; Hayward and Baker, 1968; McElligott and Melzak, 1967; Moser and Mathiesen, 1996; Thornton, 2003)").

We thank the Reviewer for these comments. We revised the text following the suggestions accordingly and removed the statements and references related to brain temperatures.